# Dual Switch in Lipid Metabolism in Cervical Epithelial Cells during Dysplasia Development Observed Using Raman Microscopy and Molecular Methods

**DOI:** 10.3390/cancers13091997

**Published:** 2021-04-21

**Authors:** Katarzyna Sitarz, Krzysztof Czamara, Joanna Bialecka, Malgorzata Klimek, Slawa Szostek, Agnieszka Kaczor

**Affiliations:** 1Chair of Microbiology, Department of Molecular Medical Microbiology, Faculty of Medicine, Jagiellonian University Medical College, 18 Czysta Street, 31-121 Krakow, Poland; katarzyna.sitarz@doctoral.uj.edu.pl; 2Faculty of Chemistry, Jagiellonian University, 2 Gronostajowa Street, 30-387 Krakow, Poland; 3Jagiellonian Centre for Experimental Therapeutics (JCET), Jagiellonian University, 14 Bobrzynskiego Street, 30-348 Krakow, Poland; krzysztof.czamara@uj.edu.pl; 4Centre of Microbiological Research and Autovaccines, 17 Slawkowska Street, 31-016 Krakow, Poland; joanna.bialecka@cbm.com.pl; 5Clinic of Radiotherapy, Maria Sklodowska-Curie Institute—Oncology Center, 11 Garncarska Street, 31-115 Krakow, Poland; malgorzata.klimek@onkologia.krakow.pl

**Keywords:** HPVhr, lipid droplets, cervical cancer, cervical dysplasia, Raman microscopy, mitochondrial DNA, methylation, *SREBF1*

## Abstract

**Simple Summary:**

Raman microscopy is an inexpensive and label-free method. The literature describes many attempts to use this method for cancer diagnosis. In this study, we used it to differentiate the lipid profile of cervical epithelial cells depending on the severity of pathological changes and the presence of HPVhr infection. Using molecular methods, we also determined the degree of methylation of the gene encoding the prolipidogenic protein SREBP1, as well as the number of copies of the mitochondrial genome in the tested samples. This multimethodological approach allowed not only to determine the differences between samples with different advancement of pathological changes, but also enabled to shed light on the molecular mechanism behind them, as well as gave hope for the possibility of using our approach for early detection of cervical dysplasia in the future.

**Abstract:**

Cellular lipid metabolism is significantly transformed during oncogenesis. To assess how dysplasia development influences lipid cellular metabolisms and what is the molecular background behind it, cervical epithelial cells of 63 patients assigned to seven groups (based on the cytological examination and HPVhr test results) were studied using a multimethodological approach including Raman microscopy and molecular methods. The consistent picture obtained studying the lipid content, cell inflammation, *SREBF1* gene methylation (hence SREBP1 inhibition) and level of mitochondrial DNA copies (indirectly the number of mitochondria) showed that changes in lipid metabolism were multidirectional. Cells from patients classified as mildly dysplastic (LSIL) exhibited a unique behavior (the highest level of inflammation and *SREBF1* methylation, the lowest lipid content and mitochondrial DNA). On the contrary, cells from severe dysplastic (HSIL) and cancer (SCC) groups showed the opposite characteristics including the lowest *SREBF1* gene methylation as well as the highest level of mitochondrial DNA and lipid cellular concentration (for HSIL/HPVhr+ and SCC groups). Following dysplastic progression, the lipid content decreases significantly (compared to the control) for mildly abnormal cells, but then increases for HSIL/HPVhr+ and SCC groups. This intriguing dual switch in lipid metabolism (reflected also in other studied parameters) on the way from normal to squamous carcinoma cells is of potential diagnostic interest.

## 1. Introduction

In 2019, cervical cancer was the second leading cause of cancer death among women in the 20–39 age group around the world [1]. Cervical cancer is caused by the sexually transmitted human papillomavirus (HPV)—it is estimated that 99% of cervical cancers develop as a result of infection with this virus [2]. HPV infection also causes a significant percentage of cancers of the mouth and throat, vulva, and anus [3,4]. However, among HPV there are over 200 types that differ in their oncogenic potential [5,6]. The most oncogenic types of HPV are 16 and 18 and these types are present in 70–75% of cervical cancers [7].

The HPV genome has six early genes (E1, E2, E4–E7), two late genes (L1, L2), and a long control region [8]. The integration of the viral genome into the host genome plays a very important role in the process of oncogenesis—the gene encoding the viral E2 protein may be damaged, which leads to an increase in the expression of the oncogenic proteins E6 and E7 [9]. The E6 protein forms a complex with the p53 protein, the product of a cell suppressor gene, which leads to its ubiquitination and degradation, creating an open path to uncontrolled cell division. In turn, the E7 protein binds to the active domains of the Rb protein, which prevents Rb from interacting with the transcription factor E2F, leading to the stimulation of cell division [10].

In the case of cervical cancer, the incidence and mortality rates can be significantly improved by extensive prevention—it is believed to be the most suitable neoplasm for primary and secondary prevention [11]. The primary prevention is vaccination against HPV [12]. The secondary prevention is cytology and HPV infection testing [13].

Lipids play a very important role in cancer, not only because the lipid signaling is crucial in processes such as invasion, migration, or metastasis [14]. In the process of carcinogenesis, the entire cellular metabolism is reprogrammed so that it is beneficial for a rapidly dividing cell [15]. Neoplastic cells activate the phosphatidylinositol 3-phosphate kinase (PI3K) pathway, which is normally activated in response to binding to cell surface receptors [16]. Activation of PI3K leads to the activation of the Akt kinase (protein kinase B), which is responsible for increasing glucose uptake, by incorporation of GLUT-1 and GLUT-4 transporters into the cell membrane [17]. Activation of Akt kinase also increases the level of glycolysis, by stimulating the activity of two glycolytic enzymes: hexokinase and phosphofructokinase [15]. Additionally, in cancer cells, the tricarboxylic acid (TCA) cycle is significantly lowered [15]. The metabolism of the cell is switched over to aerobic glycolysis, because the cell key metabolic needs of proliferating cells are nucleotides, amino acids and lipids, necessary for rapidly dividing cells and not energy as the nutrients are constantly supplied in circulating blood [15]. The change in carbohydrate metabolism is closely related to the change in lipid metabolism—activation of the Akt/Pi3K pathway causes an increase in the expression of sterol regulatory element-binding proteins (SREBPs) [18,19]. SREBPs stimulate the synthesis of two key enzymes that are involved in the formation of fatty acids: acetyl-CoA carboxylase (ACC) and fatty acid synthase (FASN) [20]. ACC is responsible for the synthesis of malonyl-CoA, while FASN catalyzes the reaction of fatty acid chain extension [20,21]. Thus, in cancer cells, due to the activation of Akt/PI3K, carbohydrate breakdown and lipid synthesis are promoted. Cancer is also usually associated with high levels of unsaturated fatty acids, important signal transducers in cancer stimulating proliferation and preventing apoptosis [22,23,24,25]. Increased lipid levels in neoplastic cells are related to increased stearoyl-CoA desaturase (SCD1) activity [25]. It is an enzyme that converts saturated fatty acids (SFAs) into monounsaturated fatty acids (MUFAs). However, this trend is reversed in human liver cancer—cancer cells have statistically lower levels of unsaturated fatty acids than healthy cells [26], although an increase in SCD levels has also been demonstrated [25].

Changes in gene expression in tumors, including genes related to cellular metabolism, are also associated with epigenetic changes [27]. DNA methylation, which is an epigenetic modification, plays a key role in gene expression [28]. Altered methylation patterns were found in all types of cancer [29]. Most often it is hypermethylation of the promoter region, which reduces gene expression, or intra-gene hypomethylation, which usually increases its expression [29].

Mitochondria play a critical role in the metabolism of cancer cells [30]. They are not only the cellular energy generators but also a biosynthesis site of compounds necessary for the cell, such as Krebs cycle intermediates. Two key but opposing processes in lipid metabolism are β-oxidation and synthesis. Both of these processes take place with the participation of mitochondria: beta-oxidation entirely in the mitochondria, herein there is also a preliminary step enabling the synthesis of lipids in the cytosol: the conversion of acetyl-CoA from pyruvate [31,32]. A key regulatory point between both processes is acetyl-CoA carboxylase (ACC), which is activated or deactivated depending on the accumulation of specific compounds in the cell [33]. Fatty acid synthesis can also be regulated by the activation and deactivation of the pyruvate dehydrogenase complex (PDHC) [34]. Many cancers also show changes in the mitochondrial DNA (mtDNA) itself, including specific mutations or a greater amount of the mitochondrial genome in the cell, which indirectly indicates a greater number of mitochondria [35,36].

Raman spectroscopy is a label-free and non-destructive method that is used to study the chemical composition of samples, including biological ones [37]. Cervical cancer cells have been studied with the use of Raman microscopy [38,39,40,41] and infrared spectroscopy [42] but they did not concentrate on changes in the lipid profile of cells.

This paper presents a multiparameter approach based on high-resolution Raman microscopy and molecular methods, particularly, quantitative real-time polymerase chain reaction (qPCR) to assess the lipid profile in cervical epithelial cells of cervical smears of high-risk HPV (HPVhr) positive or negative patients with different degrees of cervical dysplasia/cervical cancer and correlate it with the genetic background. Our unique approach enabled to demonstrate that the increased level of lipids observed in the late stages of oncogenesis was linked to the high level mtDNA copies. We have also shown that lipids unsaturation, inflammation, and a degree of CpG island methylation of the *SREBF1* gene are dysplasia dependent and, counterintuitively, the most pronounced in the LSIL group.

## 2. Materials and Methods

### 2.1. Clinical Specimens

Cervical smears were obtained from 63 women in the age of 19–76 years, living in southern Poland in the period from October 2017 to August 2020. Samples containing cervical epithelial cells were obtained from The Centre of Microbiological Research and Autovaccines, in memory of Jan Bobr and from the Department of Gynecological Oncology, National Research Institute of Oncology, Krakow Branch. The study included both women who underwent prophylactic Pap smears (samples collected in The Centre of Microbiological Research and Autovaccines; samples from 48 women) and women during their first visit to the referral Institute of Oncology (samples from 15 women). Samples at the Institute of Oncology were collected before invasive procedures and initiation of therapy (biopsies, radiochemotherapy, etc.). The cancer diagnosis was confirmed by histopathological examination by the Institute of Oncology. In our study, we included only samples of confirmed squamous cell cervical cancer.

In this work, patients have been divided into 7 groups based on the results of the cytological examination and the polymerase chain reaction (PCR) for the presence of HPVhr infection. Cervical cells were classified as normal, dysplastic (LSIL and HSIL) or cancerous according to Bethesda 2014 Classification [43]. The evaluation was performed by an experienced cytologist. Groups included patients whose smears were classified as normal, HPVhr negative (N−); normal, HPVhr positive (N+); low grade squamous epithelial lesion, HPVhr negative (LSIL−); low grade squamous epithelial lesion, HPVhr positive (LSIL+); high grade squamous epithelial lesion, HPVhr negative (HSIL−); high grade squamous epithelial lesion, HPVhr positive (HSIL+); and patients with squamous cervical carcinoma, HPVhr positive (SCC+). Full details on the size of the groups can be found in the Appendix A (Appendix A). Samples taken from each patient were divided into two parts—one for PCR testing and the other for Raman imaging. For DNA isolation for PCR, cells were frozen at −20°C until the start of the assay. For Raman imaging, freshly isolated cells were fixed using a 2.5% solution of glutaraldehyde for 4 min, then washed twice with PBS and stored in PBS in 4°C until the measurement. To carry out Raman measurements, cells were placed on a Raman substrate (CaF_2_ slides, Crystran LTD., Poole, UK). The trial was approved by the Bioethics Committee of the Jagiellonian University (23 February 2018, identification code: 1072.6120.29.2018). Written informed consent was obtained from all participants.

### 2.2. Raman Microscopy

Raman imaging of cells was performed using an Alpha 300 confocal Raman microscope (WITec, Ulm, Germany) equipped with a CCD detector (DU401A-BV-352, Andor, Belfast, UK) and a UHTS 300 spectrograph (600 grooves·mm^−1^ grating, the spectral resolution of 3 cm^−1^). A laser power of ca. 28 mW on a sample provided by a solid-state 532 nm laser source was used. To collect Raman spectra a 63× water immersion objective (Zeiss Fluor, NA = 1.0, Zeiss, Oberkochen, Germany) was applied. Raman spectra were obtained from uneven areas covering over half the area of the cell, including cell nuclei with a fragment of cytoplasm. The integration time was 0.3 s and the sampling density was 0.5 µm. At least 3 cells were measured on average for each patient. Cells that were selected for imaging: single cells, not part of an aggregate since in the aggregate the signal from neighboring cells was overlaid; lying flat on a slide as the signal from wrapped cells could be falsified; undamaged, e.g., cells with a prolapsed nucleus were rejected. Among these cells, we randomly selected cells for Raman imaging. A total of 197 cells were imaged.

Pre-treatment data processing including the procedure of cosmic rays removal and the background subtraction (polynomial of 3 degree) was performed using the WITec Project Plus 5.1 software (WITec, Ulm, Germany). To identify and separate clusters containing the cell nucleus, LDs, and cytoplasm, the obtained spectra were subjected to Cluster Analysis (CA, K-means, Manhattan distance). The Opus 7.2 software was used for the next steps of analysis: the vector normalization (spectral ranges was 1500–400 cm^−1^ and 3000–2800 cm^−1^), the averaging of spectra in the analyzed groups and calculations of the integral intensities of bands in the 3000–2830 cm^−1^ range (lipids) or at 486 cm^−1^ (glycogen), 1270 and 1300 cm^−1^ (the last two signals to calculate the lipid unsaturation ratio). The obtained results are presented in relative units.

### 2.3. HPV Testing

Detection of the presence of HPVhr infection was carried out in two stages. The first step was to perform nested PCR using two primer pairs: external MY09/MY11 and internal GP5+/GP6+ using Mastercycler Nexus × 2 from Eppendorf (Hamburg, Germany). This test is designed to detect fourteen types of HPVhr: HPV16, 18, 31, 33, 35, 39, 45, 51, 52, 56, 58, 59, 66, and 68. To visualize the test results, agarose gel electrophoresis with the addition of bromodeoxyuridine (BrdU) was performed. The second step was to perform a Cobas HPV Test (Roche, Basel Switzerland) to confirm that the detected HPV infections were associated with highly oncogenic types of the virus. Complete data on the presence of genotypes 16 and 18 in each group can be found in the Appendix A (Appendix A).

### 2.4. Counting Leukocytes in Samples

To determine the level of inflammation in specific groups, the number of leukocytes in the vaginal fornix and the cervical shield and canal were collected from the database of the Microbiological Research Center in all patients who underwent a cytological examination in 2017–2019. This number has been termed “very numerous” (leukocytes cover about 75% of the field of view), “numerous” (leukocytes cover about 50% of the field of view), “not numerous” (leukocytes cover about 25% of the field of view) and “none” (leukocytes cover 0% of the field of view) by an experienced cytologist.

### 2.5. Determination of mtDNA Copy Number

The qPCR was used to determine the amount of mitochondrial DNA in test samples relative to the amount of nuclear-encoded beta-actin gene. Two pairs of specific primers were used for the reaction: one pair for the amplification of the mitochondrial *ND1* gene (described by Warowicka et al. [36]) and the other pair for the amplification of the nuclear beta-actin gene (described by Guzik et al. [44]).

The reaction was performed on the CFX96 Touch Real-Time PCR Detection System (Bio-Rad, Hercules, CA, USA). The total volume of the PCR mixture was 10 µL. The mixture consisted of a template DNA (5 ng/reaction), 1 mM of each forward and reverse primers and 5 µL of RT HS-PCR Mix SYBR^®^A (A&A Biotechnology, Gdansk, Poland). Both genes were amplified for each sample during one thermal-cycler reaction with the same temperature profile: initial denaturation at 95 °C for 3 min, followed by 40 cycles of 95 °C for 30 s, 56 °C for 60 s, and 72 °C for 60 s. All the reactions for samples were run in duplicate. The obtained data were analyzed using double delta Ct analysis. The geometric mean of the values obtained in normal, HPVhr negative samples, in which additionally no inflammation was found, was used as the control value in the calculations.

### 2.6. Study of CpG Islands Methylation in SREBF1 Gene

To investigate the CpG island methylation of the *SREBF1* gene, bisulfite-sequencing of these genes was performed. First, sample conversion was performed with bisulfite reagent (CiTi Converter DNA Methylation Kit, A&A Biotechnology). After conversion, PCR was performed in Eppendorf Mastercycler^®^ nexus (Eppendorf). The specific primer pair described by Lou et al. [45] was used to amplify the *SREBF1* gene. The total volume of the PCR mixture was 25 µL. The mixture consisted of a template DNA (6 ng/reaction), 0.4 mM of each forward and reverse primers, 2.5 mM of MgCl_2_ and 12.5 µL of CiTi Converter MSP PCR Kit (A&A Biotechnology). The temperature profile was: initial denaturation at 95 °C for 5 min, followed by 40 cycles of 95 °C for 15 s, 58 °C for 30 s, and 72 °C for 60 s, final elongation at 72 °C for 5 min, and cooling at 10 °C. 

After the PCR reaction, the presence of the products was verified by agarose gel electrophoresis. Then, the samples were purified using the EPPiC kit (A&A Biotechnology) and subjected to Sanger sequencing.

### 2.7. Statistical Analysis

Statistical analysis was performed using the Origin 9.1 (OriginLab Corporation, Northampton, MA, USA) and STATISTICA 13.1 software (TIBCO Software Inc., Palo Alto, CA, USA). The Shapiro–Wilk test was used to check the normality of data distribution. Then, Kruskal-Wallis and U’Mann-Whitney tests were performed to assess the lipid content, glycogen content, lipid unsaturation, leucocyte count and CpG islands methylation. The Chi-square test was used to calculate differences in the level of leukocytes. A Spearman’s correlation method was used to calculate the correlation between data. Two results (both from N− group) were discarded from the cytoplasmic lipid analysis, two results (one from LSIL and one from SCC group) were discarded from unsaturation analysis and three results (from N−, LSIL− and SCC+ group) were discarded from mtDNA analysis, as they were classified as outliers based on interquartile range.

## 3. Results

### 3.1. Subcellular Distribution of Lipids in Cervical Epithelial Cells

Raman imaging was performed to determine changes in the lipid profile of cervical epithelial cells depending on the severity of pathological changes. Cells obtained from 63 patients, classified into seven groups based on the Bethesda 2014 system [43] and HPV tests as follows: N−, N+, LSIL−, LSIL+, HSIL−, HSIL+, and SCC+, were analyzed. Raman images (Figure 1A) were obtained to visualize the distribution of various components, in particular lipids, proteins, nucleic acids (DNA/RNA), and glycogen. To obtain spectra from individual subcellular structures (Figure 1B,C), the cluster analysis (KMCA, k-means, Manhattan distance) was performed. The analyzed subcellular structures are nucleus, lipid droplets (LDs) and cytoplasm—denoted in orange, blue and green, respectively. LDs were additionally divided into saturated lipids-rich—green class and unsaturated lipids-rich—purple class. LDs heterogeneity occurs between cells, there is no substantial variability inside the cells. Appendix A (Appendix A) shows the distribution of lipids in representative cells from each test group (N, LSIL, HSIL, SCC).

LDs were observed in approximately half of all imaged epithelial cells and were randomly distributed in the cytoplasm. They exhibit bands characteristic for lipids (1266, 1304, 1440, 1664, 1747 and 3006 cm^−1^) [46]. The signal at 1747 cm^−1^ arising from the C=O stretching vibration of esters indicates triacylglycerols as the main components of studied LDs. Due to the heterogeneity of observed LDs, they were divided into two groups according to their unsaturation status according to the intensity ratio of the signals at 1660 and 1440 cm^−1^ (due to the C=C stretching and CH_2_ scissoring vibrations, respectively) [46]. Moreover, observed LDs show different levels of esters. High levels of polyunsaturated fatty acids are observed in inflammation and cancer, mainly because they act as mediators in these processes [22]. When comparing the averaged signatures of cytoplasm and LDs, it is clear that bands due to lipids are present only in LDs (Figure 1B). Moreover, both of these classes also contain glycogen bands (481, 577, 858, 937, 1083, 1129, and 1340 cm^−1^) [38] but they are not present in all cells and their intensity significantly vary between cells. In the LDs of the studied cells, these components are intertwined and cannot be discriminated from each other with the use of the CA. As shown in our previous paper, the level of glycogen in cervical epithelial cells depends on the degree of dysplasia and HPVhr infection, in particular, is very low in large-nucleus HPVhr+ cells vs. HPVhr− cells, which may be associated with the increased protein metabolism [38]. The described Raman bands and their assignments have been included in the Appendix A (Appendix A).

### 3.2. The Lipid Level Is Significantly Elevated in the Cytoplasm of Cervical Epithelial Cells in the HSIL+ and SCC+ Groups

High-spatial resolution Raman microscopy enabled to estimate of lipid and glycogen contents in the cytoplasm of studied cells. A very important observation is that the lipid content of cervical epithelial cells (calculated as the integral intensity in the 2830–3000 cm^−1^ range for the whole cytoplasm with LDs/glycogen) is significantly elevated in cells of HSIL+ and SCC+ groups compared to the others (Figure 2A). A statistically significant difference in lipid levels in HSIL+ cells compared to HSIL− cells was also noticed, whereas no statistically significant differences were observed in other HPVhr+ groups vs. the respective HPVhr− ones. Complete statistical data on the lipid content in the cytoplasm have been included in the Appendix A (Appendix A). Additionally, the glycogen level anti-correlates with the lipid content in the cytoplasm of cervical epithelial cells, in particular, it is the lowest in the HSIL+ and SCC+ groups (Figure 2B), in agreement with our previous results [38]. The negative correlation of both factors is statistically significant in the N and HSIL groups (Figure 2C). 

### 3.3. The Level of Lipid Unsaturation in Cervical Epithelial Cells Correlates with the Leucocyte Levels Confirming Increased Inflammation in the LSIL Group 

The level of lipid unsaturation in LDs of cervical epithelial cells was obtained based on Raman data. Lipid unsaturation is the highest in the LSIL group and the lowest for cancer cells (SCC group, Figure 3A). Complete statistical data on the lipid unsaturation in LDs have been included in the Appendix A (Appendix A). Previously increased lipid unsaturation was directly related to inflammation due to activation of the arachidonic cycle [47,48]. 

To confirm quantitatively increased inflammation in the LSIL group, the level of leukocytes from vaginal fornix and cervical shield & canal in cervical swabs collected from 2017–2019 in The Centre of Microbiological Research and Autovaccines, including data for N, LSIL, and HSIL groups was collected and compared (Figure 3B,C). The number of leukocytes was assigned by an experienced cytologist as: very numerous, numerous, not numerous and none. The leukocyte count for both vaginal fornix and cervical shield & canal shows the following decreasing tendency in groups: LSIL > HSIL > N. The lipid unsaturation decreases in the same order confirming the conclusion regarding the increased inflammation in the LSIL group.

### 3.4. The Level of CpG Island Methylation of the SREBF1 Gene Is Statistically the Highest in LSIL Group and the Lowest in SCC Group

Another interesting correlation was found studying the CpG island methylation level of the *SREBF1* gene in the considered groups. Figure 4A,B show that the methylation level decreases in the following order: LSIL > N > HSIL > SCC. It directly correlates with the glycogen content in the cytoplasm and reversely with the lipid levels (Figure 2A,B). However, it should be noted that for most of the samples in groups N (16), HSIL (8) and SCC (9) no methylation was found. Full data on the number of methylations is provided in the Appendix A (Appendix A). Figure 4B shows the percentage of samples with at least one methylation in each group. The percentage of samples with at least one methylation are as follows: LSIL > N > HSIL > SCC. The mean number of methylations in the sample shows the same trend for the individual groups as the percentage of methylated samples (Figure 4A). 

### 3.5. The Level of Mitochondrial DNA Copies Is the Highest for the HSIL+ and SCC+ Groups

To indirectly assess whether pathological and control cells differ in the number of mitochondria, the number of copies of mitochondrial genomes was estimated. The level of mitochondrial DNA copies is highest for the HSIL+ and SCC+ groups (Figure 5). There is a tendency showing the increase in the mtDNA copy level in the presence of HPVhr infection within the N and HSIL groups, however, the statistical significance of these data was not achieved. Complete statistical data on the level of mtDNA copies have been included in the Appendix A (Appendix A).

## 4. Discussion

Cells that proliferate at high frequency, such as cancer cells, must efficiently produce biomass such as lipids, proteins, and nucleic acids [49]. The results obtained by us regarding the level of lipids in the cytoplasm are consistent with this assumption—the highest lipid levels are obtained in epithelial cervical cells are in the HSIL+ and SCC+ groups (Figure 2A). The phenomenon of de novo lipid synthesis in neoplastic cells is associated with the increased expression of the FASN and ACC enzymes [20]. FASN catalyzes the condensation of acetyl-CoA to long-chain fatty acids [20], while ACC is responsible for the formation of malonyl-CoA [21]. One of the mechanisms behind increased FASN expression is its regulation by SREBP protein [50]. In tumors, SREBPs are activated by the Akt/PI3K and mammalian target of rapamycin complex 1 (mTORC1) pathways [18,19]. The oncogenic HPV E6 and E7 proteins are among the many activators of the Akt/PI3K pathway [51,52], which may explain the increased lipid levels in the HSIL+ group compared to the HSIL− group (Figure 2A).

Contrary to the lipid content in the cytoplasm of the cervical cells, the concentration of glycogen is the lowest for the HSIL+ and SCC+ groups, and the highest for the LSIL− and LSIL+ groups (Figure 2B). Cancer cells, as well as other rapidly dividing cells, metabolize glucose by aerobic glycolysis [53]. In the past, this phenomenon was thought to be indicative of damage of mitochondria, but van der Heiden et al. [15] hypothesized that glucose catabolism is unprofitable from the point of view of rapidly proliferating cells as they need acetyl-CoA and glycolysis intermediates to produce biomass for division, which otherwise would be lost as carbon dioxide during oxidative phosphorylation. Except that, via alternative pathways to glucose oxidation, cells obtain NADPH, which is necessary for the synthesis of various macromolecules [15]. This process is also regulated by the mammalian target of rapamycin kinase (mTOR) and Akt/PI3K pathways, and it provides substrates for lipogenesis [15].

The lack of a negative correlation between the level of lipids and glycogen in the cytoplasm of neoplastic cells (Figure 2C) may seem inconsistent with the fact of increased activation of the Akt/PI3K pathway in neoplasms. It is particularly noteworthy that this correlation is disturbed by cells with zero glycogen levels in the cytoplasm. Referring to the results of Curtis et al. [54] it can be hypothesized that these are cells associated with the formation of metastases. In turn, Lee et al. [55] described that the formation of lymph node metastases is associated with the shift of lipid metabolism in cells towards the fatty acid oxidation pathway.

The highest level of lipid unsaturation of LDs was observed in the LSIL group, and the lowest in the SCC group (Figure 3A). The increase in unsaturation of fatty acids in cells is associated with carcinogenesis and inflammation [23,24,47] in literature. Studies on six different types of cancer using mass spectrometry have shown that tumors increase the level of MUFAs and are negatively correlated with the level of polyunsaturated fatty acids (PUFAs) and SFAs [56]. However, the increase in the level of lipid unsaturation is not typical of all types of cancer—an decrease in fatty acid unsaturation was also reported in the case of human liver cancer [26]. To verify if this result is associated with inflammation, we compared the leucocyte counts in the studied groups. Based on the analysis of the results of several thousand samples, we concluded that the highest level of leukocytes, both from vaginal fornix and cervical shield and canal, is present for the LSIL group (Figure 3B,C). According to Geisler et al. [57], the elevated leukocyte count is a strong predictor of vaginal and cervical inflammation. Other authors have shown that leukocytes are rarely found in the vaginal discharge of healthy women, except during menstruation [58].

To shed light on the observed variations in lipid levels in cervical cells, methylation of the *SREBF1* gene encoding SREBP1 was assessed. SREBP1 is a protein from the SREBP family, responsible for increasing the level of de novo lipogenesis and glycolysis by stimulating the expression of proteins involved in these processes [59,60]. SREBP1 also inhibits the process of beta-oxidation, increasing the concentration of malonyl-CoA (that can be used for lipid synthesis) in the mitochondrial membrane [61]. We noticed the highest level of *SREBF1* gene methylation in the LSIL samples and the lowest in the SCC samples (Figure 4A,B). It is noted in the literature that a higher degree of CpG island methylation in the selected gene fragment indicates a decrease in its expression [45]. Literature data show an association of increased SREBP1 expression with pancreatic, breast and colon cancers, as well as with a poor prognosis in these tumors [62,63,64]. Moreover, depletion of this gene expression in pancreatic cancer results in tumor growth inhibition [62].

Furthermore, we asked ourselves a question if observed changes are dependent on the number or activity of mitochondria, as various key processes in eukaryotic cells lipid metabolism take place in mitochondria [65]. Based on the measurement of mtDNA levels in the tested samples, we concluded that mitochondria are the most numerous in the HSIL+ and SCC+ groups, and the least in LSIL− and LSIL+ (Figure 5). Previously it was reported that an increase in the mitochondrial genome copy number in cervical epithelial cells is associated with dysplasia and oncogenesis [36,66]. The obtained results correlate with the amount of lipids in the cytoplasm and inversely correlate with methylation of the *SREBF1* gene (Figure 2A and Figure 4A,B). In cancer, the ACC level is upregulated, and its suppression causes a significant reduction in the viability of cancer cells [67,68,69]. Seemingly paradoxically, the key enzyme that enables lipid synthesis, PDHC, is deactivated in cancer by pyruvate dehydrogenase kinase (PDK) [70]. However, in cancer cells, another mitochondrial-active enzyme, acetyl-CoA synthetase 3 (ACSS3), is overexpressed [71]. Based on obtained data, we also put forward the statement that HPVhr infection increases the number of copies of the mitochondrial genome, but further research on a bigger cohort of patients is needed to confirm this.

The results obtained based on Raman imaging (contents of lipids and glycogen, lipid unsaturation) correlate very well with the results obtained by molecular methods (methylation of the *SREBF1* gene, the level of mitochondrial DNA) as well as leukocyte count, giving a very consistent picture of lipid changes in cervical cells developing dysplasia and, moreover, show the added value of a multimethodological approach.

The biggest limitation of our study is the relatively small number of patients in each group. It is, however, important to note that the results are backed up by several different methods that give complementary results. Other factors that have not been taken into account may also affect the results, such as the day of the menstrual cycle, past menopause, diet, physical activity, and health.

## 5. Conclusions

The increase in lipid levels in neoplastic cells has been repeatedly described in the literature. This phenomenon is related to SREBP activation through the Akt/PI3K and mTOR pathways [18,19]. Using a unique, multimethodological approach, we confirmed that this phenomenon occurs also for cervical epithelial cells classified as HSIL, moreover, is HPVhr-dependent, hence it manifests only for HPVhr positive cells. This result shows that HPVhr infection actively changes the cell epithelial metabolism making it similar to that of the cancer cell. The HSIL+ and SCC+ groups, showing the highest lipid level in the cytoplasm, demonstrate also the highest number of mitochondrial genome copies indicating the metabolic switch in these cells. The increased number of mitochondria in cells, indirectly evidenced by the number of mtDNA copies, either increases the efficiency of lipid synthesis in mitochondria of cancer-like cells or results from the inefficiency of aerobic glycolysis in energy production or both.

A decrease in the level of CpG island methylation in the analyzed fragment of the *SREBF1* gene in neoplastic cells proves a higher expression of this gene clearly associated with a global increase in the level of lipid synthesis in the cell. We also observed that the correlation of lipid and glycogen levels in the cytoplasm is negative for normal and HSIL groups, this phenomenon does not occur in LSIL and cancer cells. In the LSIL group, the phenomenon may be disturbed by inflammation. After subtracting cells with zero glycogen levels in the cytoplasm, the described relationship also occurs for neoplastic cells. Referring to the literature, it may be suspected that cancer cells with no glycogen may be involved in the metastasis process, but further research is needed to confirm this hypothesis [54,55]. If, however, these assumptions are confirmed, Raman microscopy may be useful in the rapid selection of cervical cancer patients at higher risk of metastasis based on cellular lipid and glycogen content. An interesting finding, and certainly requiring further research, is that the LDs of cervical cancer cells have a low level of lipid unsaturation contrarily to cells from the LSIL group that demonstrate the highest lipid unsaturation related to the highest inflammation as confirmed by the number of leucocytes in cervical smears.

Overall, *SREBF1* methylation and mitochondrial DNA levels are in line with observed lipid levels in cervical cells with different levels of dysplasia, showing that LSIL, normal cells and severely dysplastic/cancer cells (i.e., HSIL+ and SCC+) show very different metabolic characteristics. Of particular interest is a dual switch of the lipid metabolism observed for cells from the LSIL group (a decreased lipid level and related parameters) and HSIL+ and SCC+ groups (an increased lipid level) compared with the control cells. This phenomenon requires certainly further studies.

## Figures and Tables

**Figure 1 cancers-13-01997-f001:**
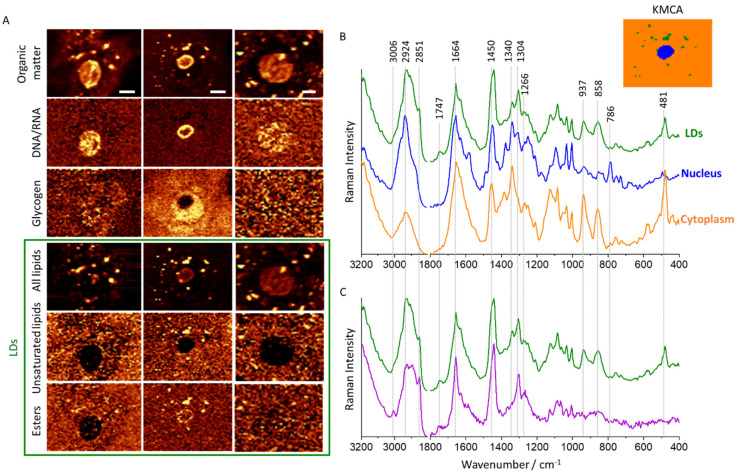
**Subcellular distribution of lipids in cervical epithelial cells**. Representative Raman images of cervical epithelial cells (N−, LSIL+, LSIL+) (**A**) obtained by integration in the regions: 2830–3030 cm^−1^ (all organic matter), 770–800 cm^−1^ (DNA & RNA), 900–955 cm^−1^ (glycogen), 2830–2900 cm^−1^ (total lipids), 2990–3020 cm^−1^ (unsaturated lipids) and 1715–1760 cm^−1^ (esters). Average Raman spectra and representative K-means cluster analysis image (**B**) showing the distribution of main classes in cervical epithelial cells: orange—cytoplasm, blue—nucleus, green—LDs. Average Raman spectra of LDs with an utmost difference of lipid unsaturation and glycogen content (**C**) of different cells indicating their heterogeneity. Scale bars equal 6 µm.

**Figure 2 cancers-13-01997-f002:**
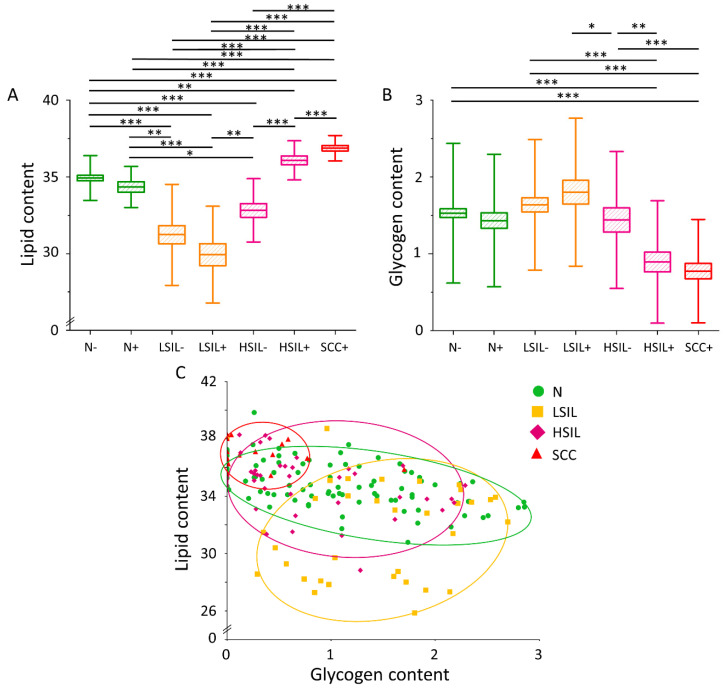
Lipid and glycogen contents in the cytoplasm of cervical epithelial cells depending on the degree of dysplasia and HPVhr infection. The comparison of the lipid (**A**) and glycogen (**B**) content in the cytoplasm of cervical epithelial cells in the studied groups obtained by calculations of the integral intensity in the 2830–3000 cm^−1^ and 458–482 cm^−1^ range, respectively. Mean values ± SEM are given as box plots: mean (horizontal line), SEM (box), SD (whiskers). Lipid content as a function of glycogen content in the cytoplasm of cervical epithelial cells (**C**) The average lipid and glycogen contents obtained by calculations of the integral intensity of the marker bands in the 2830–3000 cm^−1^ and 458–482 cm^−1^ range, respectively) in the studied groups. Each mark represents data for one cell. A significant correlation was obtained for the N (*p* < 0.001) and HSIL (*p* < 0.01) groups. * *p* < 0.05, ** *p* < 0.01, *** *p* < 0.001.

**Figure 3 cancers-13-01997-f003:**
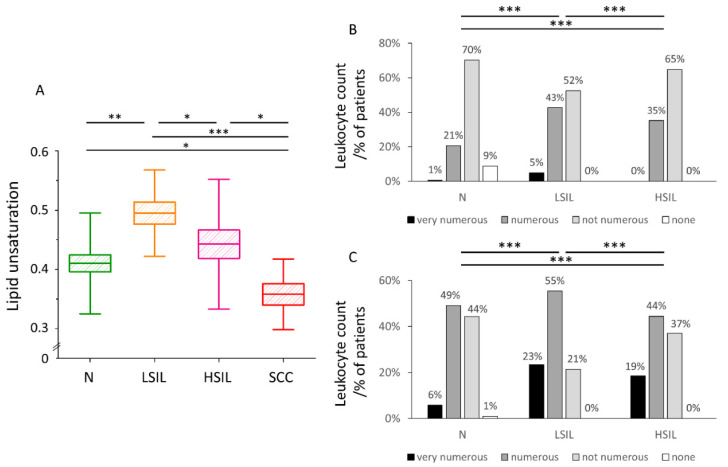
**Lipid unsaturation in LDs of cervical epithelial cells correlates with the number of leucocytes in cervical smears**. The comparison of the lipid unsaturation level in LDs of cervical epithelial cells in the studied groups obtained by calculating the ratio of the integral intensity of the bands at 1270/1300 cm^−^^1^ (**A**). Mean values ± SEM are given as box plots: mean (horizontal line), SEM (box), minimal and maximal values (whiskers). Percentage of patients grouped based on the number of leukocytes in the cervical Scheme 2017 at The Centre of Microbiological Research and Autovaccines (**B**,**C**). The comparison of the leukocyte count in the vaginal fornix (**B**) and cervical shield & canal (**C**) in the studied groups: N, LSIL, and HSIL. The leukocytes in each sample were assessed by the cytologist as: very numerous (leukocytes cover about 75% of the field of view), numerous (leukocytes cover about 50% of the field of view), not numerous (leukocytes cover about 25% of the field of view) and none (leukocytes cover 0% of the field of view). * *p* < 0.05, ** *p* < 0.01, *** *p* < 0.001.

**Figure 4 cancers-13-01997-f004:**
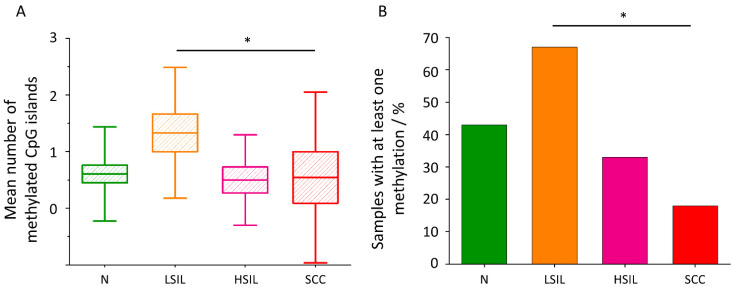
**CpG island methylation in the *SREBF1* gene.** The comparison of the mean number of methylated CpG islands of the studied fragment of *SREBF1* gene in the studied groups (**A**). Mean values ± SEM are given as box plots: mean (horizontal line), SEM (box), SD (whiskers). * *p* < 0.05. Percentage of samples with at least one methylation in the *SREBF1* gene. * *p* < 0.05 (**B**).

**Figure 5 cancers-13-01997-f005:**
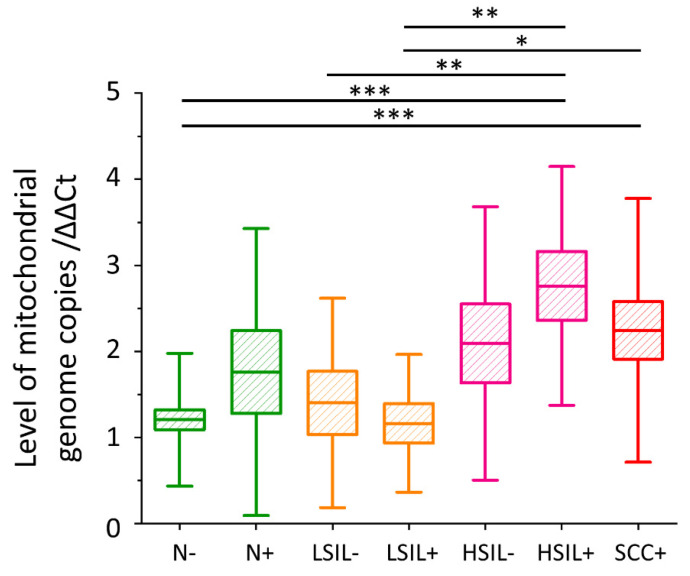
**The level of mitochondrial DNA copies in cervical epithelial cells.** The comparison of the level of mtDNA copies in cervical epithelial cells in studied groups obtained by calculating a Ct value from the Ct value of the ND1 gene relative to the Ct value of the beta-actin gene. Mean values ± SEM are given as box plots: mean (horizontal line), SEM (box), SD (whiskers). * *p* < 0.05, ** *p* < 0.01, *** *p* < 0.001.

## Data Availability

The data is archived at the Department of Molecular Medical Microbiology, Chair of Microbiology, Jagiellonian University Medical College, 18 Czysta Street, 31-121 Krakow, Poland.

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
