# Peer review of "Dual Switch in Lipid Metabolism in Cervical Epithelial Cells during Dysplasia Development Observed Using Raman Microscopy and Molecular Methods"

_cancers, 2021, doi:10.3390/cancers13091997_

Round 1

Reviewer 1 Report

The manuscript presents the data to explain the lipid profile in cervical epithelial cells of HPV-positive/negative patients with different degrees of dysplasia by utilizing Raman microscopy and molecular methods. It is a well-written manuscript, and the following are the comments that will improve the manuscript's quality.

Major comments.

In the Raman microscopic experiments, authors should explain how cells were classified as normal, dysplastic, or cancer cells? As cell samples from cervical smears contain a mixture of the cells, for example, SCC samples have 1 to 10 % of cancer cells, and the remaining are normal (histologically, based on the nucleus to cytoplasmic ratio)

- Following the previous comment, In Figure 1- the authors mentioned representative images of N-, LSIL+, LSIL+,

How were the cells selected for the imaging? It could be explained in detail in the material and methods section.

- It would give readers more clarity if authors could include the Raman images for SCC and their lipid distribution in the cells. 

- How many cells were imaged? It could be mentioned in the Methods and material section.

- For HPV testing, the Authors mentioned that the first test was to identify 14 types of highly oncogenic HPV. The second step was to perform a Roche Cobas HPV test to confirm that the detected HPV infection was associated with highly oncogenic types of virus. If all the HPV positive sample presented in the manuscript were high risk (HPVhr), then the author should mention it consistently throughout the manuscript, including supplementary material.

- Were their HPV-18 and 16 specifically identified, or the primers were generic to all the 14 types of high-risk HPV? 

- In the discussion section, authors should include the paragraph mentioning the Raman microscopic finding and how it is correlated with molecular data (Lipid/glycogen content, inflammation and CpG island methylation)? If not, then explaining why?

In the discussion section, authors can also acknowledge the heterogeneity of samples. For example, in Squamous cell carcinomas (SCC), the cancer cells' percentage ranges from 1-10% cancer cells. The remaining are normal cells. It should be explained in detail.  

Minor comments.

- In Figure 3B, C and Figure 4 B, the standard deviation bar is missing. Please consider including it.

- It would be great if authors could also show the sample distribution using box-blots(https://www.nature.com/articles/nmeth.2813.pdf?origin=ppub)   

- Figure 1: Kindly check for the typo error 'N-, LSIL+, LSIL+'

- Kindly consider changing the title, as it does not give an accurate representation of the manuscript.

- It would be great if authors could include a paragraph explaining the limitations of the study.

- Consider including the Raman band annotation in tabular form along with the reference in supplementary section. 

Author Response

The manuscript presents the data to explain the lipid profile in cervical epithelial cells of HPV-positive/negative patients with different degrees of dysplasia by utilizing Raman microscopy and molecular methods. It is a well-written manuscript, and the following are the comments that will improve the manuscript's quality.

Major comments.

  1. In the Raman microscopic experiments, authors should explain how cells were classified as normal, dysplastic, or cancer cells? As cell samples from cervical smears contain a mixture of the cells, for example, SCC samples have 1 to 10 % of cancer cells, and the remaining are normal (histologically, based on the nucleus to cytoplasmic ratio)

Thank you for this comment. The cervical epithelial cells were classified into normal, dysplastic, and neoplastic on the basis of conventional cytology and histopathological examination (SCC).

At the same time, two smears were taken from each woman, one for Pap smear tests and the other, which was collected in saline and properly secured and stored until Pap smear results were obtained. Raman imaging cells were randomly selected.

Changes in the manuscript are as follows:

2.1 (lines 145-147): Smear cells were classified as normal, dysplastic (LSIL and HSIL) or cancerous according to Bethesda 2014 Classification [43]. The evaluation was performed by an experienced cytologist.

  1. Following the previous comment, In Figure 1- the authors mentioned representative images of N-, LSIL+, LSIL+,

It was not our intention to present in the figure the heterogeneity of lipid distribution between the groups. The drawing was intended to show the heterogeneity of cells in the entire studied population.

  1. How were the cells selected for the imaging? It could be explained in detail in the material and methods section.

This is indeed an important point and we added the information about it. The method of selecting cells for imaging has been added to the "Materials and methods" section. Random cell selection for Raman imaging was important because, in the case of molecular studies, there is no option to select cells with a particular morphology, and we wanted consistent results for imaging and molecular testing.

Changes in the manuscript are as follows:

2.2. (lines 172-176): Cells that were selected for imaging: single cells, not part of an aggregate since in the aggregate the signal from neighboring cells was overlaid; lying flat on a slide as the signal from wrapped cells could be falsified; undamaged, e.g. cells with a prolapsed nucleus were rejected. Among these cells, we randomly selected cells for Raman imaging.

  1. It would give readers more clarity if authors could include the Raman images for SCC and their lipid distribution in the cells.

Thank you for this idea. A figure showing the distribution of lipids in N, LSIL, HSIL and SCC groups was added to the Supplement (Figure S1).

Changes in the manuscript are as follows:

3.1 (lines 266-267): Fig. S1 (Supplementary Materials) shows the distribution of lipids in representative cells from each test group (N, LSIL, HSIL, SCC).

  1. How many cells were imaged? It could be mentioned in the Methods and material section.

The information was added to the "Materials and methods" section.

Changes in the manuscript are as follows:

2.2 (line 180): A total of 197 cells were imaged.

  1. For HPV testing, the Authors mentioned that the first test was to identify 14 types of highly oncogenic HPV. The second step was to perform a Roche Cobas HPV test to confirm that the detected HPV infection was associated with highly oncogenic types of virus. If all the HPV positive sample presented in the manuscript were high risk (HPVhr), then the author should mention it consistently throughout the manuscript, including supplementary material.

Thank you for this comment. Changes have been added in appropriate places to both the manuscript and the supplement.

  1. Were their HPV-18 and 16 specifically identified, or the primers were generic to all the 14 types of high-risk HPV? 

Thank you for the idea of supplementing this data. The types HPV-16 and HPV-18 have been specially identified. Information on the percentage of samples infected with these types among the HPVhr positive samples in each group is provided in Table S2 in the supplement.

Changes in the manuscript are as follows:

2.3 (lines 199-201): Complete data on the presence of genotypes 16 and 18 in each group can be found in the Supplementary Materials (Table S2).

  1. In the discussion section, authors should include the paragraph mentioning the Raman microscopic finding and how it is correlated with molecular data (Lipid/glycogen content, inflammation and CpG island methylation)? If not, then explaining why?

Thank you for this remark, indeed maybe it was not crystal clear that the lipid/glycogen content and lipid unsaturation were obtained based on Raman microscopy.

Therefore, at the beginning of the paragraph 3.2 and 3.3. it was underlined that these results were obtained based on Raman data for cytoplasm and also, the paragraph, summarizing the correlation between the results was added in the discussion section.

Changes in the manuscript are as follows:

3.2 (lines 299-300):  High-spatial resolution Raman microscopy enabled to estimate lipid and glycogen content in the cytoplasm of studied cells.

3.3 (lines 326-327): The level of lipid unsaturation in in LDs of cervical epithelial cells was obtained based on Raman data.

Discussion (lines 464-468): The results obtained on the basis of Raman imaging (content of lipids and glycogen, lipid unsaturation) correlate very well with the results obtained by molecular methods (methylation of the SREBF1 gene, the level of mitochondrial DNA) as well as leukocyte count, giving a very consistent picture of lipid changes in cervical cells developing dysplasia and, moreover, show the added value of a multimethodological approach.  

  1. In the discussion section, authors can also acknowledge the heterogeneity of samples. For example, in Squamous cell carcinomas (SCC), the cancer cells' percentage ranges from 1-10% cancer cells. The remaining are normal cells. It should be explained in detail.

As already mentioned in point 1, unfortunately, we were not able to determine the percentage of pathological cells in the samples. However, we are grateful for this valuable comment and will consider it in future experiments.

Minor comments.

  1. In Figure 3B, C and Figure 4 B, the standard deviation bar is missing. Please consider including it.

Including standard deviation bars in the above-mentioned charts is impossible due to the fact that these charts present percentage data. We considered presenting these data in the form of a table, however, they found the charts to be more attractive to read.

  1. It would be great if authors could also show the sample distribution using box-blots (https://www.nature.com/articles/nmeth.2813.pdf?origin=ppub)   

Thank you for this remark. Charts have been changed to box-blots, with the exception of charts showing percentages, which cannot be converted to boxes.

  1. Figure 1: Kindly check for the typo error 'N-, LSIL+, LSIL+'

As mentioned above, this is not a typo error. The graph is intended to show the overall cell heterogeneity. However, as also noted above, Fig. S1 has been added to the supplement, which shows the distribution of lipids in the representative cells of the individual groups.

  1. Kindly consider changing the title, as it does not give an accurate representation of the manuscript.

Thank you for this comment. The title was changed to: "Dual switch in lipid metabolism in cervical epithelial cells during dysplasia development observed using Raman microscopy and molecular methods".

  1. It would be great if authors could include a paragraph explaining the limitations of the study.

Limitations of the study have been added to the discussion section.

Changes in the manuscript are as follows:

Discussion (lines 469-473): The biggest limitation of our study is a relatively small number of patients in each group. It is, however, important to note that the results are backed up by several different methods that give complementary results. Other factors that have not been taken into account may also affect the results, such as the phase of the menstrual cycle, past menopause, diet, physical activity, and health.

  1. Consider including the Raman band annotation in tabular form along with the reference in supplementary section. 

Thank you for the advice. We have included the table in the Supplement (Table S3).

Changes in the manuscript are as follows:

3.1 (lines 294-296): The described Raman bands and their assignments have been included in the Supplementary Materials (Table S3).

Reviewer 2 Report

Sitarz et al. report on metabolomic analysis of cervical epithelial cells from control, dysplasic and cancer patients exhibiting HPV infection or not. Total lipid content, proportion of unsaturation, level of inflammation, amount of mitochondrial DNA and level of DNA methylation of the SREBF1 gene are presented.

The work is solid, rather well presented and the proper statistical tests are performed. On that last note, although statistical repeats of cells from each patient increase statistical power, the number of patients in disease groups remains small. Which may pose a bias to the conclusions of the study. Although I understand it is not easy to increase the number of patients, I think this should be more clearly written in the discussion section.

Author Response

Sitarz et al. report on metabolomic analysis of cervical epithelial cells from control, dysplasic and cancer patients exhibiting HPV infection or not. Total lipid content, proportion of unsaturation, level of inflammation, amount of mitochondrial DNA and level of DNA methylation of the SREBF1 gene are presented.

The work is solid, rather well presented and the proper statistical tests are performed. On that last note, although statistical repeats of cells from each patient increase statistical power, the number of patients in disease groups remains small. Which may pose a bias to the conclusions of the study. Although I understand it is not easy to increase the number of patients, I think this should be more clearly written in the discussion section.

Thank you for a nice review. We do agree that the number of samples is limited, however the results were supported by several methods giving compatible results. We added a comment about it in the discussion section, according to your suggestion.

Discussion (lines 469-473): The biggest limitation of our study is a relatively small number of patients in each group. It is, however, important to note that the results are backed up by several different methods that give complementary results. Other factors that have not been taken into account may also affect the results, such as the phase of the menstrual cycle, past menopause, diet, physical activity, and health.

Round 2

Reviewer 1 Report

Thank you for incorporating the suggested changes.